# Yu–Shiba–Rusinov screening of spins in double quantum dots

K. Grove-Rasmussen [1], G. Steffensen[1], A. Jellinggaard[1], M.H. Madsen[1], R. Žitko[2,3], J. Paaske[1] & J. Nygård[1]

A magnetic impurity coupled to a superconductor gives rise to a Yu–Shiba–Rusinov (YSR) state inside the superconducting energy gap. With increasing exchange coupling the excitation energy of this state eventually crosses zero and the system switches to a YSR ground state with bound quasiparticles screening the impurity spin by $\hbar/2$. Here we explore indium arsenide (InAs) nanowire double quantum dots tunnel coupled to a superconductor and demonstrate YSR screening of spin-1/2 and spin-1 states. Gating the double dot through nine different charge states, we show that the honeycomb pattern of zero-bias conductance peaks, archetypal of double dots coupled to normal leads, is replaced by lines of zero-energy YSR states. These enclose regions of YSR-screened dot spins displaying distinctive spectral features, and their characteristic shape and topology change markedly with tunnel coupling strengths. We find excellent agreement with a simple zero-bandwidth approximation, and with numerical renormalization group calculations for the two-orbital Anderson model.

[1] Center for Quantum Devices & Nano-Science Center, Niels Bohr Institute, University of Copenhagen, Universitetsparken 5, 2100 Copenhagen Ø, Denmark. [2] Faculty of Mathematics and Physics, University of Ljubljana, Jadranska 19, SI-1000 Ljubljana, Slovenia. [3] Jozef Stefan Institute, Jamova 39, SI-1000 Ljubljana, Slovenia. Correspondence and requests for materials should be addressed to K.G-R. (email: k_grove@fys.ku.dk)

Yu–Shiba–Rusinov (YSR) states[1–3] can be imaged in a direct manner by scanning-tunneling spectroscy of magnetic adatoms on the surface of a superconductor[4]. Using superconducting tips, high-resolution bias spectroscopy of multiple sub-gap peaks reveals an impressive amount of atomistic details like higher angular momentum scattering channels, crystal-field splitting and magnetic anisotropy[4–11]. In general, however, it can be an arduous task to model the complex pattern of sub-gap states[6,7,9], let alone to calculate their precise influence on the conductance[10].

In contrast, the "atomic physics" of Coulomb blockaded quantum dots (QDs) is simple. Changing the gate voltage, subsequent levels are filled one-by-one and the different charge states alternate in spin, or Kramers degeneracies for dots with spin–orbit coupling, between singlet and doublet. With normal metal leads, charge states with spin-1/2 exhibit zero-bias Kondo resonances at temperatures below the Kondo temperature, $T \ll T_K$, reflecting a Kondo-screened singlet ground state (GS). If the leads are superconducting with a large BCS gap, $\Delta \gg k_B T_K$, this resonance is quenched and the GS recovers its doublet degeneracy. The system now displays a YSR singlet excitation close to the gap edge, which can be lowered in energy by increasing the $k_B T_K/\Delta$[12–17]. Close to $k_B T_K \approx 0.3\Delta$ it crosses zero and becomes the YSR-screened singlet GS[9,11,18–20], which eventually crosses over to a Kondo singlet at $k_B T_K \gg \Delta$.

YSR states were first discussed in the context of gapless superconductivity arising in the presence of randomly distributed paramagnetic impurities[1–3]. However, the ability to assemble spins into dimers, chains, and lattices, has recently prompted the exciting idea of engineering YSR molecules[20,21], YSR sub-gap topological superconductors and spiral magnetic states[4,22–24]. QDs have the advantage of being tunable via electrical gates, and this plays an important role in recent proposals for topological superconductivity in systems of coupled QDs[25–27].

Here we utilize this electrical control to manipulate YSR states in a double quantum dot (DQD) formed in an InAs nanowire. Using multiple finger gates to tune the total DQD spin and the interdot coupling, we demonstrate control of the YSR phase diagram, including electrical tuning between YSR singlets, and a novel YSR doublet arising from the screening of an excited spin triplet.

## Results

**Device and model.** A scanning electron micrograph of an actual device (Device A) is shown in Fig. 1a, where bottom gates are used to define a normal (N)-DQD-superconductor (S) structure[17]. The corresponding schematic is shown in Fig. 1b, where plunger gates labeled $g_N$ and $g_S$ control left (QD$_N$) and right (QD$_S$) quantum dot, respectively, while an auxiliary gate, $g_d$, tunes the interdot tunneling barrier. The essential physics of this system can be understood in terms of a simple zero-bandwidth (ZBW) model in which the superconductor is modeled by a single quasiparticle coupled directly to an orbital in QD$_S$ via $t_S$ and indirectly to QD$_N$ through $t_d$. A normal metal electrode with weak coupling $t_N$ and correspondingly low Kondo temperature to the left dot is used to probe the DQD-S system. Figure 1c shows the corresponding energy diagram in the regime of dominating on-site charging energies. In the Supplementary Note 3 we compare this model to numerical renormalization group (NRG) calculations to establish its reliability as a quantitative tool.

In Fig. 2a, b, we reproduce the well-known sub-gap state behavior for a single dot coupled to a superconductor within the ZBW model. The panels show excitation energy as a function of the dimensionless gate voltage $\tilde{n}_S$ (corresponding to the noninteracting

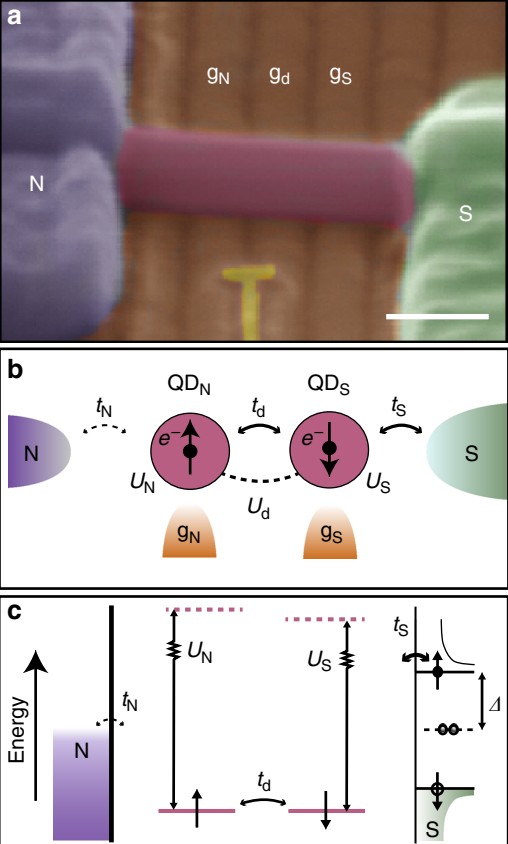

**Fig. 1** Device layout. **a** False colored scanning electron micrograph of Device A showing a normal (N)—InAs nanowire—superconductor (S) device, where a double dot is defined by appropriate voltages on the bottom gates. The yellow floating gate intended for charge sensing by a nearby quantum dot is not used. The scale bar corresponds to 100 nm. **b** Schematic of a double quantum dot coupled to a normal and a superconducting electrode with couplings $t_N$ and $t_S$, respectively. The electrostatic potentials on the two dots are controlled by gates $g_N$ and $g_S$, respectively, while the tunnel coupling $t_d$ between the dots is tuned by gate electrode $g_d$. Similarly, the charging energies of quantum dot QD$_S$, QD$_N$ and the mutual charging energy are given by $U_N$, $U_S$ and $U_d$, respectively. **c** Energy diagram of a normal—double quantum dot—superconductor device with charging energies larger than the superconducting gap, $U_i \gg \Delta$, $i = $ N, S

average occupation of QD$_S$) for weak and strong $t_S$. As expected the sub-gap excitations cross (do not cross) zero energy for weak (strong) coupling. The GS of the system for odd occupancy is thus a doublet or a YSR singlet (screened spin)[13,17,18].

In Fig. 2f–i we extend the ZBW model to a DQD (finite $t_d$) and calculate stability diagrams for increasing $t_S$. For weak coupling the characteristic honeycomb pattern is observed similar to DQD in the normal state[28–30]. However, as $t_S$ increases, entirely new types of stability diagrams emerge. In Fig. 2g, the pattern resembles two mirrored arcs, where the lack of zero-energy excitations as a function of $\tilde{n}_S$ for even occupation of QD$_N$ is due to a doublet to singlet transition in QD$_S$ (see Fig. 2a, b). Moreover, as the coupling increases even further, the GS in the $(\tilde{n}_N, \tilde{n}_S) = (11)$ region becomes a YSR doublet altering the stability diagram to vertically shaped rectangular regions.

To understand this behavior, we show the states of the system in the (11) region in Fig. 2c. Two electrons in the DQD may form either a singlet $\mathcal{S}_{11}$ or a triplet $\mathcal{T}_{11}$ state with energy splitting $J_d$.

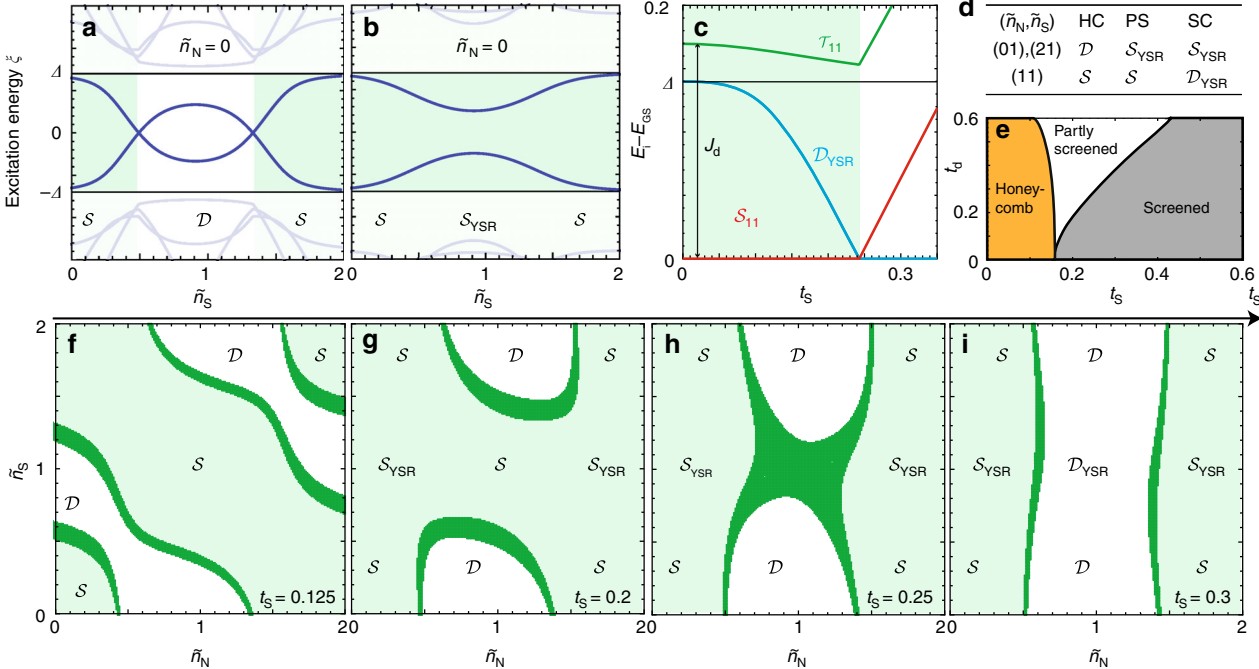

**Fig. 2** YSR phase diagrams. **a, b** Two distinct YSR sub-gap spectra vs. occupation $\tilde{n}_S$ of $QD_S$ for $\tilde{n}_N = 0$ (single dot case). For weak and strong coupling to the superconductor, the doublet $\mathcal{D}$ (**a**) and YSR singlet $\mathcal{S}_{YSR}$ (**b**) is the ground state for odd occupancy, respectively. **c** ZBW model calculation in the (11) charge state, with singlet $\mathcal{S}_{11}$ and triplet $\mathcal{T}_{11}$ separated by the interdot exchange energy $J_d$. The triplet is YSR-screened and gives rise to a sub-gap YSR doublet $\mathcal{D}_{YSR}$, which becomes the ground state (GS) for large enough $t_S$. **d, e** Double-dot YSR phase diagram hosting three regimes (**e**): honeycomb (HC), partly screened (PS) and screened (SC). The ground states in the (01) and (21), and the (11) charge regions for the three regimes are shown in **d**. **f–i** Stability diagrams for increasing $t_S$ show the transitions between honeycomb (**f**), partly screened (**g**) and screened (**i**) regimes. The ground states are explicitly stated for the YSR-screened states, while trivial singlet and doublet states may cover several charge sectors. A singlet-doublet energy splitting less than 0.015 meV (i.e. close to degenerate) defines the dark green region. For **h** $\mathcal{S}_{11}$ and $\mathcal{D}_{YSR}$ are almost degenerate corresponding to a transition in (11) which is unique to the DQD-S system. Parameters (in meV) used in **d–i** are extracted from experimental data (see Supplementary Note 2, Device A): $U_N = 2.5$, $U_S = 0.8$, $U_d = 0.1$, $t_d = 0.27$, $t_S = 0.22$, $\Delta = 0.14$

Due to the superconductor, a third state may also exist in the gap. In analogy with the QD-S system, where a doublet state may be screened to a YSR singlet, the triplet state may be screened to form a YSR doublet (called $\mathcal{D}_{YSR}$)[11]. The energy of $\mathcal{S}_{11}$ and $\mathcal{D}_{YSR}$ versus coupling is plotted in Fig. 2c, and the latter eventually becomes the GS at strong coupling.

The relevant GSs and corresponding regimes with two GS transitions in the ($t_S$ versus $t_d$)-plane of the DQD-S system is shown in Fig. 2d, e. The first occurs when the system transitions from a honeycomb pattern to the case where the spin in $QD_S$ is screened. The latter regime we call partly screened (PS) since only some of the charge states are affected. The second transition happens when the $\mathcal{D}_{YSR}$ in (11) becomes the GS. This regime we name screened (SC) since all possible screened states are GSs of the system. We emphasize that the names of these regimes do not describe the degree of screening of the individual spin states, e.g., in the screened regime the triplet giving rise to $\mathcal{D}_{YSR}$ is underscreened, while the doublet spin giving rise to $\mathcal{S}_{YSR}$ is completely screened (see Fig. 2d). The ($t_S$, $t_d$) position of the regime boundaries are dependent on choice of parameters, but the overall behavior stays the same. For instance, for larger $U_S$, the transitions move toward larger $t_S$ as one would expect.

**Measurements**. With the qualitative behavior of this system in place, we explore the different regimes experimentally. The honeycomb regime is presented in the Supplementary Note 2 (Device B), while below we focus on the stronger coupled

regimes. Figure 3a shows linear conductance versus plunger gates for a two-orbital DQD shell (i.e., one spin-degenerate level in each dot). A pattern of two arcs is observed, resembling the PS regime. To verify that the conductance resonances originate from sub-gap states, gate traces for different fillings of the two dots are measured. Figure 3c–e trace out the filling of electrons in $QD_N$ along the red arrows in Fig. 3a, keeping the electron number in $QD_S$ constant. The sub-gap spectroscopy plots c,e for even filling of $QD_S$ show similar behavior, differing from d with odd occupation. When fixing (sweeping) the occupation of $QD_N$ ($QD_S$), the qualitative behavior is switched (Fig. 3f–h corresponding to green arrows in Fig. 3a). For even occupancy in $QD_N$ (f,h) no zero-bias crossing is observed, while the opposite is true for odd occupancy (g). In particular Fig. 3d, g are interesting, since they involve the (11) charge state region. In contrast to single dot systems, the singlet GS shows different behavior whether tuning the electrochemical potential of the dot close to the superconductor or the normal lead, i.e., concave and convex excitation behavior versus gate voltage in the (11) state. The experimental data clearly confirm that the resonances in the stability diagram originate from sub-gap excitations. The stability diagram generated by our DQD-S model for realistic parameters reproducing the experimental behavior is shown in Fig. 3b, and corresponding gate traces for fixed occupations are shown in Fig. 3i–n. The qualitative agreement between theory and experiment is striking and even subtleties like the asymmetry of the sub-gap resonance splitting in j (see arrows) are reproduced.

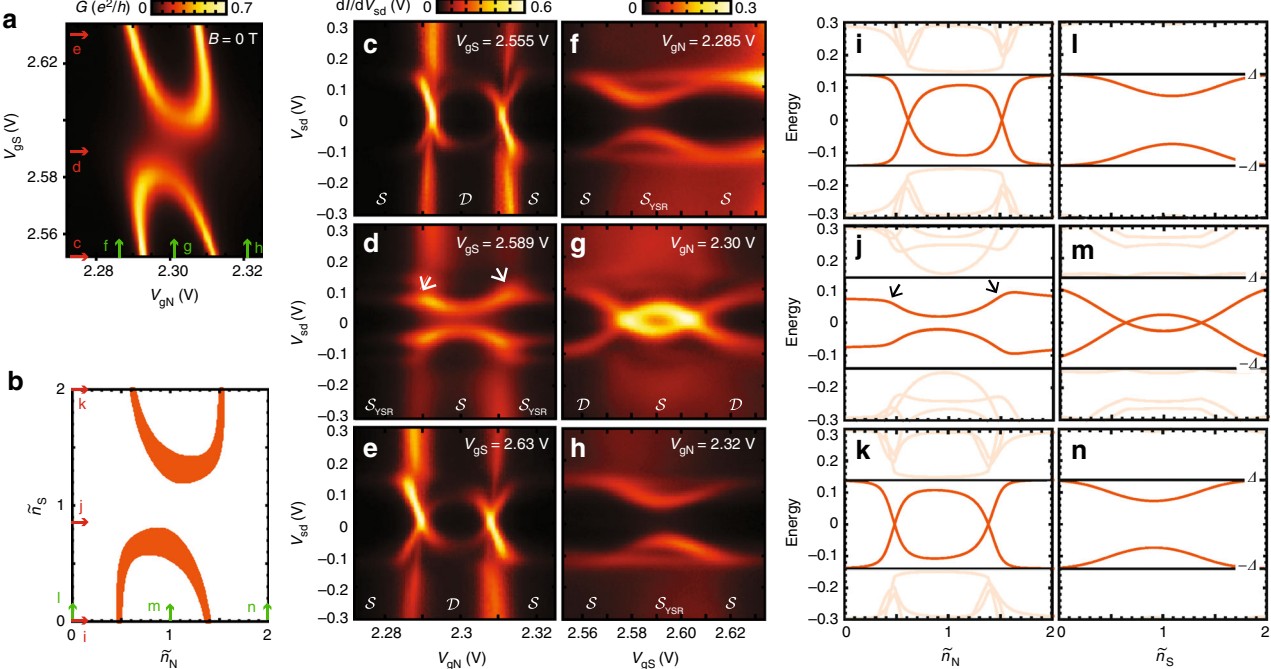

**Fig. 3** Sub-gap states in the partly screened regime. **a** Stability diagram showing linear N-DQD-S conductance vs. plunger gates at $T = 30$ mK. **b** DQD-S zero-bandwidth model reproducing the experimental behavior in **a** for intermediate coupling $t_S$ to S. Orange regions marking ground state transitions have singlet-doublet splitting less than 0.015 meV. **c–e** Bias spectroscopy of sub-gap states vs. $QD_N$ occupation, sweeping $V_{g_N}$ as indicated by red arrows in **a**. **f–h** Sub-gap spectroscopy vs. $QD_S$ occupation sweeping along green arrows in **a**. All plots show clear sub-gap resonances consistent with the ground states (doublet $\mathcal{D}$ or singlet $\mathcal{S}$) for different gate voltages indicated below. **i–n** Zero-bandwidth model calculation of sub-gap excitations corresponding to experimental plots **c–h**. Parameters used in **b, i–n** are (meV): $U_N = 2.5$, $U_S = 0.8$, $U_D = 0.1$, $t_d = 0.27$, $t_S = 0.22$, and $\Delta = 0.14$. Additional data related to the partly screened regime and how the above parameters are extracted from experimental data can be found in the Supplementary Note 2, Device A

The transition between different YSR states can also be driven by changing the singlet-triplet ($\mathcal{S}_{11}$-$\mathcal{T}_{11}$) splitting by tuning $t_d$ (cf. Fig. 2e). In Fig. 4e–h we show calculated diagrams for $t_d$ in a parameter range, where the GS in the (11) charge state transitions from $\mathcal{S}_{11}$ (h) to $\mathcal{D}_{YSR}$ (e), i.e. from double-arc, to vertical-lines diagrams. The corresponding measured stability diagrams for the two orbitals analyzed in Fig. 3 are shown in Fig. 4a–d, where the gate voltage between the two dots are tuned to more negative values (decreasing $t_d$). The effect of this tuning qualitatively follows the expectation of the model: a transition from $\mathcal{S}_{11}$ to $\mathcal{D}_{YSR}$ in the screened regime where all spin states are YSR screened.

The gate-dispersion of sub-gap excitations also shows good overall correspondence between ZBW modeling and experiment. We measured the sub-gap spectra in the screened regime along the red and green arrows in c and a. The first case c is almost at the transition, where the singlet and doublet states are degenerate in (11). Figure 4i, j shows sub-gap states versus $V_{gN}$ and $V_{gS}$, respectively, with a zero-bias peak at $V_{gS} = 2.62$ V reflecting a degeneracy at this value of $t_d$. The corresponding ZBW modeling in Fig. 4k, l (for $t_d = 0.25$ meV) places the system just barely in the screened regime with a $\mathcal{D}_{YSR}$ (11) GS and a nearby $\mathcal{S}_{11}$ excitation dispersing very much like in the measurement. A $\mathcal{T}_{11}$ triplet state is predicted inside the gap, and should be accessible from the $\mathcal{D}_{YSR}$ GS. As demonstrated in the Supplementary Note 3, this is confirmed by more accurate NRG calculations, which however reveal a strong suppression of spectral weight on this state, explaining why it may be difficult to observe in experiment. For even lower $t_d$, case a, Fig. 4m–p again show good overall correspondence between experiment and theory, except for the triplet state, which should be weak, and in this case hardly

resolved within the linewidth broadening in the data. Future experiments with hard gap superconductors or improved resolution may eventually lead to capability to detect even such low-weight spectral features. A detailed discussion of nonlinear conductance and broadening of the YSR sub-gap spectra is provided in the Supplementary Note 4. In particular, the electron–hole (e–h) asymmetry of the sub-gap resonance amplitude in, e.g., Fig. 3f is due to relaxation from the sub-gap state to quasiparticles above the gap (i.e. in the case of no relaxation the sub-gap resonance amplitude is expected to be e–h symmetric).

## Methods

**Fabrication**. The devices are made by defining bottom gate Au/Ti (12/5 nm) electrodes (pitch 55 nm) on a silicon substrate capped with 500 nm $SiO_2$ followed by atomic layer deposition of $3 \times 8$ nm $HfO_2$. InAs nanowires (70 nm in diameter) appropriately aligned on bottom gate structures are contacted by Au/Ti (90/5 nm) normal and Al/Ti (95/5 nm) superconducting electrodes separated by ~ 350 nm[17]. The superconducting film has a critical field of around 85 mT.

**Measurements techniques**. The samples are mounted in an Oxford Instruments Triton 200 dilution refrigerator with base temperature of around 30 mK and are measured with standard lockin techniques. For the data (Figs. 3 and 4) shown in the PS regime, the voltages on the gates define a double dot potential with values (V) $V_{g1} = 0$, $V_{g2} = -1.3$, $V_{g3} = 2.3$, $V_{g4} = -0.3$, $V_{g5} = 2.65$, $V_{g6} = -0.3$, and $V_{g7} = 0.3$. Here the gate numbers correspond to gates from left to right in Fig. 1a. Gates 3, 4, and 5 are thus the left plunger, the tunnel barrier and right plunger gates, which are tuned within some range of the values stated.

**Data availability**. The data presented above can be found at the following https://sid.erda.dk/public/archives/ec32617f4b179826cb9343ce46c50b11/published-archive.html.

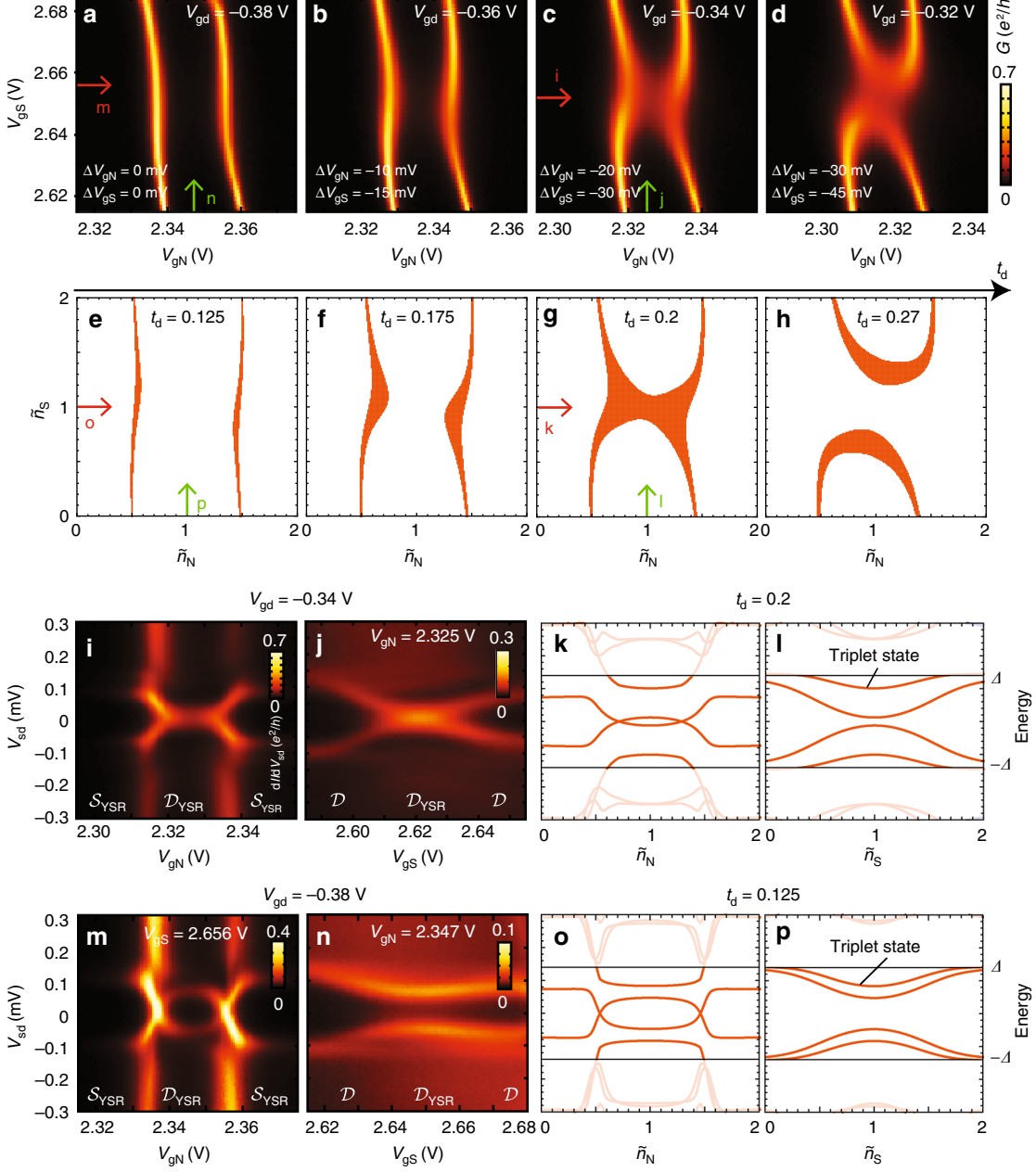

**Fig. 4** Tuning interdot coupling $t_d$. **a–d** Stability diagrams for different voltages on the tuning gate $g_d$ while compensating on plunger gates ($T = 30$ mK). The plot in **d** is analyzed in Fig. 3 and the voltage on $g_d$ is decreased in steps of $-20$ mV from **d** to **a** (compensating in steps of 10 and $-15$ mV on $g_N$ and $g_S$, respectively). **e–h**, Stability diagrams generated by the zero-bandwidth model for different $t_d$ (in meV), qualitatively reproducing the experimental behavior in **a–d**. Orange regions marking ground state transition have singlet-doublet splitting less than 0.015 meV. **i**, **j**, **m**, **n** Bias spectroscopy of sub-gap states vs. individual plunger gates swept along red, and green arrows in **c** and **a**. All plots show clear sub-gap resonances consistent with the ground states (doublet $\mathcal{D}$ or singlet $\mathcal{S}$) for different gate voltages indicated below. **k**, **l**, **o**, **p** Zero-bandwidth model calculation of sub-gap excitations for $t_d = 0.2$ meV and $t_d = 0.125$ meV corresponding to experimental plots **i**, **j**, **m**, **n**. The triplet excitation has very low spectral weight and therefore does not show up in the measured bias spectroscopy. Parameters are fixed to the same values as in Fig. 3

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

## Acknowledgements

We would like to thank M. C. Hels and C. B. Sørensen. Research was supported by the Center for Quantum Devices, The Danish National Research Foundation, Carlsberg Foundation, The Independent Research Fund Denmark (Natural Sciences), the FP7 FET-Open SE2ND project and the Slovenian Research Agency (ARRS) under Program P1-0044 and J1-7259.

## Author contributions

K.G.-R. and A.J. performed the measurements, A.J. fabricated the devices and M.H.M. grew the nanowires. K.G.-R., A.J., and J.N. designed the double quantum dot experiments, G.S., K.G.-R., and J.P. made the ZBW analysis, G.S. and J.P. the conductance asymmetry calculations, and R.Z. provided the NRG analysis. K.G.-R., G.S., J.P., R.Z., and J.N. participated in discussions, analysis and wrote the paper.

## Additional information

**Competing interests:** The authors declare no competing interests.

