## [Peer Review File · Nature Communications]

Reviewers' comments:

Reviewer #1 (Remarks to the Author):

This is an experimental-theoretical paper dealing with transport properties of a double quantum dot coupled to normal and superconducting electrodes. The authors focus on the so-called Shiba/Andreev states, which form inside the superconducting energy gap. These states have recently attracted a considerable attention of the mesoscopic physics community due to e.g. Majorana physics, Cooper pair tunneling/splitting, etc. In this respect the paper presents an interesting experimental extension of existing studies, which is well corroborated by theoretical models, including ZBW (which agrees very well with the data, to my surprise) and more accurate NRG calculations. While I think that this study may deserve publication in Nature Communications, before making the decision, I would like the authors to consider the following points:

I am not convinced to the notion of "Yu-Shiba-Rusinov screening". As known, the presence of magnetic impurity in SC results in bound states - called Yu-Shiba-Rusinov states. (Similar states can be also realized in QD setups, and can be probed by Andreev reflection spectroscopy, hence, in this context the name: Andreev bound states.) These states form inside the SC energy gap. Depending on the ratio of Δ and TK the system exhibits then a phase transition. When $TK > \Delta$, the spin is screened by SC quasiparticles and the singlet Kondo state forms. The Kondo effect however develops due to the usual spin-flip screening processes. Maybe I missed something, but the phrase YSR screening is not clear to me. Could you please comment on that?

The authors write that they demonstrate the screening of spin-1/2 and spin-1 states of the DQD. However, I think this is an over-interpretation. The triplet spin state is never truly screened, since it's not the ground state of the system. Instead the authors find a partially screened doublet state. However, as the authors correctly remark, the existence of a similar state was already predicted. From this point of view, the authors should emphasize more what is clearly new in their investigations.

The authors identify three different phases and write that the system shows two ground state transitions. I found it a little bit confusing. For fixed gates (fixed number of electrons on DQD), when tuning the couplings there is only one transition. In charge sector (even, 1) the transition is between D and S_YSR, while in sector (1,1) between S and D_YSR. It would be good to clarify this point. Similarly, a little bit confusing can be the phrase "fully screened phase". I would naively think that the spins of DQD are fully screened there, but it turns out that only one of the spins is screened while the second one is not. What the authors mean is that the states with evenly occupied dot N and singly occupied dot S are fully screened, while the state (1,1) is only partially screened. I think I got the intention of distinguishing between different behaviors in the phase diagram, but I found it a bit misleading.

In Fig. 2, it would increase the readability of the paper if the authors could somehow indicate different ground states, i.e. S_YSR and D_YSR, in the second row of panels.

The role of the normal lead should be discussed in the main text, in particular, what is the coupling to normal lead and associated TK ? From the schematic (Fig. 1b) one can get an impression that the system is symmetric.

I suggest the authors extend the discussion of energy scales when studying the phase diagrams in Fig. 2. It can be seen that the transition between S_11 and D_YSR occurs precisely for $t_S = t_d$. The choice of parameters in Fig. 2 is also not given. I guess this comes from experimental estimates(?) Since there is a large asymmetry between the dots, it would be good to comment on

how crucial this asymmetry is in the studied behavior.

The authors give the value of $\Gamma \approx 0.3 \Delta$ for the phase transition. How was this estimated?

It would be good to indicate the energy U_d in Fig. 1c (or in schematic, part b).

Reviewer #2 (Remarks to the Author):

The manuscript "Yu-Shiba-Rusinov screening of spins in double quantum dots" reports careful exploration of Yu-Shiba-Rusinov (YSR) states in nanowire quantum dots coupled to a superconductor. The manuscript combines theory and experiment to provide detailed understanding of the ground state phase diagram in this highly tunable system. The manuscript is well written, contains beautiful data of extremely high quality and has very clear explanations of the theory.

Understanding the physics of YSR states in individual atoms, molecules as well as single and double quantum dots is important step towards realization of engineered YSR systems. In this context, data reported in the manuscript is novel, timely and important as it demonstrates state of the art control of the YSR ground state phase diagram in highly tunable InAs double quantum dots. For these reasons, I strongly recommend this manuscript for publication in your journal. Having said that, I only have one comment which should be addressing prior to publication.

One of the mysteries in this field is the relaxation which leads to electron-hole asymmetry in the tunneling spectra and is characterized by the parameter (Γ_r). So far there were several speculations of possible origin of this effect, which is highly important for data interpretation both in quantum dots and magnetic atoms. Does the system investigated here provide more insights in the nature of this relaxation? In particular, is the relaxation rate independent of external parameters or temperature? While authors give estimate of the parameter in supplementary information, they barely mention relaxation process in the main text. Commenting about the relaxation effects in this system with its connection to the electron hole asymmetry would be useful to educate readers.

Another small point is that some of the abbreviations are not defined in the main text. One particular example is NRG. This should be corrected in the final version.

Reviewer #3 (Remarks to the Author):

In their work, Grove-Rasmussen et al. show evidence of Yu-Shiba-Rusinov (YSR) states in a double quantum dot contacted to one superconducting lead. These states were already well studied in a single quantum dot, and the current work extends it towards a more complex system, a double quantum dot, which offers the opportunity to investigate a configuration where the singlet-triplet splitting can be tuned as a function of the interdot coupling.

The main novelty offered by this configuration as compared to the single quantum dot is the possibility to investigate YSR state forming from the spin-polarized triplet ($S=1$) state, which would be at much larger energy in the case of single quantum dot, and the possibility to form YSR state at even total filling factor. Indeed this configuration seems to be demonstrated in Fig. 4(i-j) and 4(m-n), which demonstrate experimentally the original situation depicted in Fig. 2(i) from the model.

There is however a major question arising from the configuration shown in Fig. 4(a,m,n). It is not clear how the authors distinguish the situation where (a) the YSR state is formed with the $S=1$ triplet state of the double quantum dot (the expected new result) or (b) the YSR state forms only with the $S=1/2$ state in QD_N, while QD_S is strongly coupled to the S lead, and V_g acting mainly on the state of QD_N though the cross-capacitance (this would be an artefact, and a more trivial result). Do the authors have an independent proof of the $S=1$ nature of this state? An interesting complementary analysis should be the analysis of this configuration with the superconducting lead in the normal state. Another interesting analysis would be to investigate the influence of a magnetic field on the spin states (even with the superconducting lead in the normal state).

In addition to the major novelty described above, additional original results include (1) the influence of the filling of both quantum dots (close to the normal contact and close to the superconducting contacts) and (2) the influence of their coupling on the subgap states (YSR states). Fig. 3 shows clearly the alternating singlet and doublet states obtained while changing the filling of the quantum dots. As already mentioned before, the results of Fig. 4 are ambiguous as there is no independent proof that, when the inter-dot coupling is reduced, the involved states are still the double quantum dot states, and not simply the single quantum dot state (the ones in QD_N).

In conclusion, the work presented by Grove-Rasmussen et al. could be clearly original, and of strong interest for the scientific community interested in the coupling of spin states with superconductors. With the debate on the observation of Majorana fermions in similar systems, the current work would be timely and of wide interest more generally for the physicist community. However the demonstration of the major new result, i.e. the new YSR state obtained by coupling the $S=1$ state of the double quantum dot with a superconductor, still requires additional experimental proofs before I can recommend the paper for publication to Nature Communications.

First we would like to thank the referees for the thorough read of our manuscript and the suggestions for improving the manuscript. Below we address the points raised in the reports:

Reviewer #1:

1. The Kondo effect however develops due to the usual spin-flip screening processes. Maybe I missed something, but the phrase YSR screening is not clear to me. Could you please comment on that?

We use the term YSR screening to indicate that the phenomenon is about YSR states and the screening of the spin due to coupling to gapped quasiparticles in the leads. The referee is correct that this screening happens when the exchange coupling to the lead, here parametrized by TK , becomes sufficiently large. Increasing TK from zero, the YSR singlet (initially comprising a spin $\frac{1}{2}$ impurity and a single quasiparticle) moves from the gap-edge down through zero energy, after which it becomes the ground state. Increasing TK further, adds more and more quasiparticles to the 'screening cloud', and as such this singlet ground state is a precursor of the Kondo singlet described well by Nozières Fermi-liquid model. Yet, the BCS gap remains, and the lowest lying (doublet) excitation above this Fermi liquid ground state is visible as a sharp bound state inside the gap, until it merges with the continuum at $TK \gg \Delta$. Technically speaking, a certain set of Feynman diagrams are summed to arrive at the YSR singlet and another set of diagrams to arrive at the Kondo singlet. As the BCS gap is closed, the former must of course lose importance to the latter and the YSR screening cloud becomes identical to the Kondo cloud. To our knowledge, the detailed distinction between the two, and any possible observable way of distinguishing them, has not been worked out, but as long as the gap is observed one has not yet arrived at the Kondo Fermi liquid fixed point, and the notion of YSR screening remains a more accurate description of what goes on. We have now inserted a sentence on top of 2nd column of page 1, "... , which eventually crosses over to a Kondo singlet at $k_B T_K \gg \Delta$.", which should help the reader to understand that the former is a precursor of the latter.

We also note that the notion of YSR screening can be generalized to the situation with no Kondo effect. Such is the case of the original works by Yu, Shiba and Rusinov, where the screening (reduction of the local moment from S to $S-1/2$) occurs through spin-dependent elastic scattering (no spin flips!). Another example is provided by quantum spin models with magnetic anisotropy terms, when sub-gap states with reduced S appear in the gap even when the Kondo effect is quenched by anisotropy. Thus YSR screened state could be defined as any many-body state with the impurity local moment reduced due to exchange coupling to the superconductor (whichever the exact processes involved).

2. The authors identify three different phases and write that the system shows two ground state transitions. I found it a little bit confusing. For fixed gates (fixed number of electrons on DQD), when tuning the couplings there is only one transition. In charge sector (even, 1) the transition is between D and S_{YSR} , while in sector (1,1) between S and D_{YSR} . It would be good to clarify this point. Similarly, a little bit confusing can be the phrase "fully screened phase". I would naively think that the spins of DQD are fully screened there, but it turns out that only one of the spins is screened while the second one is not. What the authors mean is that the states with evenly occupied dot N and singly occupied dot S are fully screened, while the state (1,1) is only partially screened. I think I got the intention of distinguishing between different behaviors in the phase diagram, but I found it a bit misleading.

The referee has understood the regimes correctly and the names of the regimes were chosen to describe whether only part of (partially screened) or all (fully screened) screened spin states are the ground state in the double dot diagram as the referee points out. We do, however, understand that it may lead to

confusion. It is not easy to introduce a short terminology describing both charge state regions at the same time (underscreened regime would for instance also be incorrect for the “fully screened” regime, since the spin of the 10 charge state is fully screened). We have now changed the terminology to partly screened and screened. This is still not completely unambiguous, but in our opinion better than neutral terms such as regime II, III or intermediately and strongly screened. Furthermore, we inserted the following sentence to stress the interpretation of the regimes. “We emphasize that the names of these regimes do not describe the degree of screening of the individual spin states in the honeycomb diagram (all states are maximally screened by spin-1/2), e.g., in the screened regime the triplet giving rise to D_{YSR} is underscreened, while the doublet spin giving rise to S_{YSR} is completely screened (see table 2.1d)”.

3. In Fig. 2, it would increase the readability of the paper if the authors could somehow indicate different ground states, i.e. S_{YSR} and D_{YSR} , in the second row of panels.

We have now added the labels for the screened and trivial ground states in the honeycomb diagrams to make it easier to read off from the figure. Also the singlet and doublet states shown in Fig. 4 have been specified.

4. The role of the normal lead should be discussed in the main text, in particular, what is the coupling to normal lead and associated TK? From the schematic (Fig. 1b) one can get an impression that the system is symmetric.

In the data presented, the normal lead is coupled weakly to QD_N . We have changed the schematic by adding a thick tunnel barrier (1c), thin dashed arrow for t_N (1b,c) and smaller normal lead (1b) to indicate that the normal lead is a weakly coupled probe. A sentence explicitly stating this is inserted in the main text. “A normal metal electrode with weak coupling t_N and correspondingly low Kondo temperature to the left dot is used to probe the DQD-S system.”

To further elaborate on this point. For other data, not presented here, with the normal lead coupled more strongly to the QD_N (i.e. $T_{K_normal_side} > T$), we observe a Kondo ridge/no Kondo ridge for odd/even occupancy in the normal dot coexisting with YSR states. This indicates that T_K related to the normal electrode is small ($T_{K_normal_side} < T$) in the data shown, since no Kondo ridge is observed.

5. I suggest the authors extend the discussion of energy scales when studying the phase diagrams in Fig. 2. It can be seen that the transition between S_{11} and D_{YSR} occurs precisely for $t_S = t_d$. The choice of parameters in Fig. 2 is also not given. I guess this comes from experimental estimates(?) Since there is a large asymmetry between the dots, it would be good to comment on how crucial this asymmetry is in the studied behavior.

We have added the parameters to the caption in Fig. 2, which stems from the experimental estimate given in the Supplementary Information (now also referred to in the figure caption). The transition between S_{11} and D_{YSR} in t_d vs t_S plane does depend on the parameters, but the overall behavior is the same. For larger U_S , the transition moves to larger t_S as expected. To illustrate the change in diagrams, an overview of stability diagram versus the t_d and t_S is inserted in the Fig. 1(a-b) below. Each sub-plot shows n_N vs n_S and the color scale indicates singlet, or doublet ground state (blue or orange). Fig. 1(a) has the same parameters as in Fig. 2 of the article, while Fig. 1(b) is plotted with equal charging energy. Identifying the transition from partially to fully screened (dashed lines), it is seen that the

transition depends on the exact parameters, in this case U_S . We have added the following sentence to the manuscript. "The position of the phase boundary is dependent on choice of parameters, but the overall behavior stays the same. For instance, for larger U_S , the transition moves towards larger t_S as one would expect."

Fig 1(a) Couplings t_s (x-axis) vs t_d (y-axis) with stability diagrams (n_N vs n_S) in steps of 0.1 meV. Lower left corner $(t_s, t_d) = (0,0)$. Upper right corner $(t_s, t_d) = (0.6, 0.6)$ meV. Parameters are given in Fig. 2 of the article. Blue/red colors corresponds to singlet/doublet ground state. The PS-S regime transition is indicated by dashed line.

Fig 1(b) Couplings t_s (x-axis) vs t_d (y-axis) with stability diagrams (n_N vs n_S) in steps of 0.1 meV. Same parameters as in (a), except that U_S is increased from 0.8 \rightarrow 2.5 meV = U_N . The transitions between regimes HC-PS and PS-S are seen to move to larger t_s , but the overall diagram stays the same. The PS-S regime transition is indicated by dashed line.

6. The authors give the value of TK $\approx 0.3 \Delta$ for the phase transition. How was this estimated?

The value of the transition in the introduction was given to indicate that different regimes can be reached with a single quantum dot. This was not extracted from the data. We inserted a reference supporting this statement: J. Bauer, A. Oguri, and A. C. Hewson, J. Phys. Condens. Matter 19, 486211 (2007).

7. It would be good to indicate the energy U_d in Fig. 1c (or in schematic, part b).

The energy U_d as well as U_N and U_S have now been indicated on Fig. 1b.

Reviewer #2:

1. One of the mysteries in this field is the relaxation which leads to electron-hole asymmetry in the tunneling spectra and is characterized by the parameter (Γ_r). So far there were several speculations of possible origin of this effect, which is highly important for data interpretation both in quantum dots and magnetic atoms. Does the system investigated here provide more insights in the nature of this relaxation? In particular, is the relaxation rate independent of external parameters or temperature?

We agree with the referee that the relaxation time in these system is extremely interesting. We do unfortunately not have enough data to make a strong statement about the relaxation time, but show that an estimate can be deduced from data. The origin of the relaxation is so far not clear to us, but thorough studies of the YSR peak height asymmetry versus temperature and other parameters are one of the topics we would like to explore more in these devices.

2. While authors give estimate of the parameter in supplementary information, they barely mention relaxation process in the main text. Commenting about the relaxation effects in this system with its connection to the electron hole asymmetry would be useful to educate readers.

We have inserted a sentence referring to the analysis on the asymmetry and relaxation in the main text: "In particular, the electron-hole (e-h) asymmetry of the sub-gap resonance amplitude in e.g. Fig 3(f) is due to relaxation from the sub-gap state to quasiparticles above the gap (i.e. in case of no relaxation the sub-gap resonance amplitude is expected to be e-h symmetric)."

3. Another small point is that some of the abbreviations are not defined in the main text. One particular example is NRG. This should be corrected in the final version.

We have written out or indicated the following abbreviations. Indium Arsenide (InAs), quantum dot (QD), numerical renormalization group (NRG). We have kept BCS, SiO₂ and HfO₂ in the short form.

Reviewer #3:

Before addressing the individual questions below, we would like to state that the analysis of the experimental data gives compelling evidence to support the interpretation of a well-defined DQD device. Furthermore, the extracted model parameters are robust, and the model reproduces the totality of the sub-gap spectral features.

1. There is however a major question arising from the configuration shown in Fig. 4(a,m,n). It is not clear how the authors distinguish the situation where (a) the YSR states is formed with the $S=1$ triplet state of the double quantum dot (the expected new result) or (b) the YSR state forms only with the $S=1/2$ state in QD_N, while QD_S is strongly coupled to the S lead, and V_g acting mainly on the state of QD_N though the cross-capacitance (this would be an artefact, and a more trivial result).

In the (11) two-electron state, the energy eigenstates are spin singlet or triplet. The precise spatial form of the wavefunction of course depends on t_d . If t_d was turned off, the wave function would have two well-separated peaks, consistent with two uncoupled spin doublets (one in each dot). In that case, it would certainly be the QD_S spin which is screened. However, for finite t_d (finite conductance through the device), the two-electron states must be sorted as singlet or triplet with a certain energy splitting and the only spin around for YSR screening is therefore $S=1$. The two-electron + one-quasiparticle YSR (under-)screened state has $S=1/2$ (D_{YSR}), and most likely the wavefunction for this spin doublet will have more weight on the N-side, depending on the magnitude of t_d . In our asymmetric setup (N-DQD-S), this kind of asymmetry in the wave function is always to be expected, but this doesn't change the fact, that it is an $S=1$ state being screened. For sufficiently low t_d , the system effectively behaves like a single dot.

2. Do the authors have an independent proof of the $S=1$ nature of this state? An interesting complementary analysis should be the analysis of this configuration with the superconducting lead in the normal state. Another interesting analysis would be to investigate the influence of a magnetic field on the spin states (even with the superconducting lead in the normal state).

An independent proof would require normal-state inelastic cotunneling spectroscopy resolving the Zeeman splitting of the excited triplet, which unfortunately we do not have. Even with such spectroscopy, however, a non-ambiguous Zeeman splitting may sometime be difficult to identify, depending on the relations between the singlet-triplet splitting, the tunnel couplings and the spin-orbit coupling. Such additional evidence would of course provide an even stronger indication of the $S=1$ state (and the $S=1/2$ states for that matter), but we stress that the current interpretation is consistent with the expected behavior for the gate voltages applied to the double dot.

3. In addition to the major novelty described above, additional original results include (1) the influence of the filling of both quantum dots (close to the normal contact and close to the superconducting contacts) and (2) the influence of their coupling on the subgap states (YSR states). Fig. 3 shows clearly the alternating singlet and doublet states obtained while changing the filling of the quantum dots. As already mentioned before, the results of Fig. 4 are ambiguous as there is no independent proof that, when the inter-dot coupling is reduced, the involved states are still the double quantum dot states, and not simply the single quantum dot state (the ones in QD_N).

We cannot completely exclude that the center barrier gate will merge the two dots, but this behavior would be opposite to the expected behavior for applying negative gate voltage, i.e. the application of more negative voltage to the center gate g_d is not expected to change the double dot into a single dot.

To further analyze the double-dot nature of our data and presence of QD_S, Fig. 2 below shows stability diagrams in the superconducting and normal state ($B=140$ mT) for the shells A-F presented in Fig S2 of the supplementary material, corresponding to the settings of Fig. 4(a-b). The three lower shells (D-F) show similar transition to the screened (fully screened) regime as reported in Fig. 4. The two horizontal resonances are crossing the vertical lines (e.g. see arrows in (d)). These we assign to QD_S. In the normal state the resonances are generally broadened and two sharper (shells A-C) and one broad well coupled (shells D-F) resonance along the horizontal lines indicating the filling of QD_S.

Fig. 2: Stability diagram of conductance (log scale) vs V_{gN} and V_{gS} of the normal and superconducting state. A broadening of the resonances is observed in the superconducting state. The left and right black ovals correspond to Fig. 4a and 4b, respectively.

REVIEWERS' COMMENTS:

Reviewer #1 (Remarks to the Author):

I have read the revised manuscript very carefully. The authors have responded to all my concerns and improved the paper accordingly. I think the paper can be published in its present form.

Reviewer #3 (Remarks to the Author):

The questions raised by my first referral report were correctly addressed. While one of the main issues, which is an independent demonstration of the $S=1$ spin state in the double quantum dot, could not fully be answered, the strong indication given by the comparison with simulations is convincing. I thus recommend the publication of the article in Nature Communications.